# Facilitators and Barriers to HPV Self-Sampling as a Cervical Cancer Screening Option among Women Living with HIV in Rural Uganda

**DOI:** 10.3390/ijerph20116004

**Published:** 2023-05-30

**Authors:** Agnes Nyabigambo, Roy William Mayega, Khumbulani Hlongwana, Themba Geoffrey Ginindza

**Affiliations:** 1Department of Public Health Medicine, School of Nursing and Public Health, University of KwaZulu-Natal, Durban 4000, South Africa; rmayega@musph.ac.ug (R.W.M.); hlongwanak@ukzn.ac.za (K.H.); ginindza@ukzn.ac.za (T.G.G.); 2Department of Community Health and Behavioural Sciences, School of Public Health, Makerere University, Kampala P.O. Box 7062, Uganda; 3Health Economics and HIV/AIDS Division (HEARD), University of KwaZulu-Natal, Durban 4000, South Africa; 4Department of Epidemiology and Biostatistics, School of Public Health, Makerere University, Kampala P.O. Box 7062, Uganda; 5Cancer and Infectious Diseases Epidemiology Research Unit, School of Nursing and Public Health, University of KwaZulu-Natal, Durban 4000, South Africa

**Keywords:** HPV, facilitators, barriers, cervical cancer, HPV self-sampling, home-based, clinic-based, women living with HIV

## Abstract

**Background**: There is a paucity of studies exploring women living with HIV’s (WLWH) experiences relating to human papillomavirus (HPV) self-sampling as cervical cancer (CC) screening approach, either at the clinic or at the home setting, using qualitative methods. Our study explored facilitators and barriers to HPV self-sampling as a CC screening approach among human immunodeficiency virus (HIV)-infected women, as supported by the new WHO guidelines of using the HPV test as a screening modality. **Methods:** The study was guided by the health promotion model (HPM), which helps individuals achieve higher levels of well-being. The phenomenology design was used to explore the deeper facilitators and barriers of women regarding self-sampling, either at home or in clinical settings, at Luweero District Hospital, Uganda. The in-depth interview (IDI) guide was translated from English to Luganda. Qualitative data analysis was guided by content analysis techniques. The transcripts were coded in   NVivo 20.7.0. The coded text was used to generate categories of analytically meaningful data that guided the formation of themes, the interpretation of results, and the final write-up. **Results:** WLWH were motivated to screen for HPV using the clinic-based approach because of perceived early diagnosis and treatment, visualization of the cervix, and free service, while reduced distance, privacy and the smooth sample collection kit were motivators for the home-based approach. A barrier that cut across the two HPV self-sampling approaches was a lack of knowledge about HPV. The barriers to clinic-based HPV self-sampling screening included lack of privacy, perceived painful procedures for visual inception under acetic acid (VIA), and fear of finding the disease. Stigma and discrimination were reported as the major barriers to the home-based HPV self-sampling approach. The major reasons why some WLWH refused to screen were fear of finding the disease, stress, and financial disruptions related to being diagnosed with CC disease. **Conclusions:** Therefore, early diagnosis for HPV and CC enhances clinic-based HPV self-sampling, while privacy enhances the home-based HPV self-sampling approach. However, fear of finding a disease and the lack of knowledge of HPV and CC hinders HPV self-sampling. Finally, designing pre- and post-testing counselling programs in HIV care is likely to increase the demand for HPV self-sampling.

## 1. Introduction

Cervical cancer (CC) is the most common cancer among Ugandan women [1]. Despite the incidence of CC being as high as 54.8 per 100,000 women per year in Uganda, there is generally a poor uptake of screening services among women [2]. Cervical cancer disease is mainly caused by the high-risk types of human papillomavirus (HPV), such as HPV 16 and HPV 18 [3]. Persistent infection with HPV plays a key role in the development of cervical intraepithelial neoplasia and invasive cervical cancer [4]. HPV is a sexually transmitted infection, and the prevalence of HPV 16/18 in the general Ugandan population is high at 3.6% [5].

The prevalence of human immunodeficiency virus (HIV) remains high at 7.6% among women in reproductive health [6]. Women living with HIV (WLWH) represent the most at-risk population for acquiring HPV and are more susceptible to persistent infection with more than one strain of this high-risk HPV, primarily due to their reduced immunity function, consequently making them prone to CC [7,8]. The World Health Organization (WHO) recommends HPV testing and CC screening as secondary prevention measures [9]. Consequently, the Ugandan guidelines recommend annual screening for WLWH from ages 25 to 49 years [10]. The new WHO guidelines for cervical cancer prevention were updated in 2021 and recommend HPV testing as the primary screening test to direct the test and treatment approaches. The visual inspection of the cervix with acetic acid (VIA) was recommended as a triage test for women who test HPV-positive to guide the ablative or excisional treatment [9].

Previous studies indicated negative experiences among WLWH in relation to provider-led CC screening approaches, and these mainly included negative attitudes by health workers towards CC screening [11] and unfriendly health workers [12,13,14,15]. Worse still, primary healthcare providers tended to miss symptoms or hardly suspect cancer in the early stages, thus leading many to develop and present with the advanced, untreatable disease at a later stage [13,16]. Women also expressed community and personal embarrassment associated with provider-led screening procedures [17] and perceived low service quality [18]. On the other hand, studies also demonstrated negative experiences relating to patient-led CC screening approaches, especially HPV self-sampling, and these included limited knowledge about HPV and poverty that hindered them from accessing emerging screening programs [12,13,14,15].

However, there is a paucity of studies exploring WLWH’s experiences relating to HPV self-sampling as a CC screening approach, either at the clinic or in the home setting, using qualitative methods [19]. The ability for women to self-collect HPV samples would potentially contribute to the decongestion of health facilities while improving cervical cancer screening coverage rates, as shown in the feasibility study conducted in northern Tanzania [20]. Self-sampling for HPV testing, as an alternative to conventional speculum-based sampling, is acceptable for women of screening age [21].

Our study explored facilitators and barriers to HPV self-sampling as a CC screening approach among HIV-infected women, as supported by the new WHO guidelines on using the HPV test as a screening modality. The study was conducted among women living with HIV in a rural setting who had access to HIV care. However, HPV self-sampling was relatively a new approach within this setting, where women are at risk of HPV infection due to multiple sexual partnerships. Therefore, understanding the dynamics of clinic-based vs home-based HPV self-sampling approaches was relevant, timely, significant, and interesting to explore. Hence, this study could be conducted in any HIV clinic supporting the integration of CC screening services with HIV care.

## 2. Materials and Methods

### 2.1. Theory

The study was guided by the health promotion model (HPM) [22], which helps individuals achieve higher levels of well-being. It also inspires health workers to provide positive resources that support patients to achieve desired behavior-specific changes. The goal of HPM is not only to help patients prevent illness through their behavior but also to assess ways in which a person can pursue better health or ideal health. Therefore, this theory provided the framework through which women’s facilitators, barriers, preferences, perceptions, and intentionality relating to the HPV self-sampling screening approach, either at home or in clinical settings, could be understood.

### 2.2. Research Design

The phenomenology design was used to explore the deeper facilitators and barriers of women regarding self-sampling, either at home or in clinical settings, to elicit the barriers and facilitators to HPV screening in the Luweero District Hospital, Uganda. The paradigms underpinning the interpretations of the results included new feminist approaches and neoliberalism [23]. Since there was no clear policy or a cervical cancer national prevention program, neoliberalism [23] and women empowerment concerning HPV self-sampling were key aspects that were explored.

In this study, women were selected from the randomized, controlled trial, where women were blinded and assigned either clinic- or home-based HPV self-sampling. The vignettes technique was used to educate women on HPV self-sampling. The demonstration focused on the use of the HPV sample collection kit and how it could be used at the clinic or at home. The midwife educated WLWH on HPV self-sampling and monitored them. The community linkages person (CLP) educated the women on the HPV self-sampling kit and self-sampling procedures. WLWH collected and returned the sample to the CLP. The clinic-based HPV DNA self-sampling approach was compared to the home-based HPV DNA screening approach to obtain a deeper understanding of the uptake and preferences of self-sampling in the clinical vs the home setting. In this study, the Qvintip® sampling device was used for sample collection by WLWH. It had an oval and smooth tip, which was preferred by WLWH, compared to the brush that was the standard in use by the HIV clinic.

Information was gathered from key informants (KIIs) using in-depth interview (IDIs) guides. The KIIs included a program officer in the division of community health in the MoH, a cancer expert involved in cervical cancer at the Uganda Cancer Institute (UCI), and 4 health workers (HWs) at the Luweero District Hospital. At the district level, we interviewed the HIV clinic manager in charge of CC screening and the HIV clinic community linkage person. We interviewed 24 WLWH, all aged 25–49 years (Table 1).

### 2.3. Trustworthiness of the Study

The in-depth interview (IDI) guide was translated from English to Luganda, and the quality of the translation was assured through back-translation using a professional translator. Subsequently, the tools were pretested to eliminate ambiguities in the guides before the actual data collection. Despite the 3 research assistants (RAs) being experienced in conducting qualitative research, they were given additional study-specific training encapsulating the study aim, objectives, procedures, and ethical issues. Each interview took about 45 min to 1 h to be completed. Verbatim quotes were used to support the findings. The information gathered were related to the facilitators and barriers to the clinic- or home-based HPV self-sampling screening approaches. The key informants expressed their views on the experiences of CC screening approaches in the HIV clinic. At the inception, interview rapport was established, and women had the autonomy to decide whether to participate, freely express their views, or decline participation. The study was conducted and supervised by a team that included an expert on qualitative study methods who guided the study methods, as well as the analysis and interpretation of the findings.

### 2.4. Data Analysis

The recordings were transcribed in the local language (Luganda) and later translated into English by a professional translator whose native language was Luganda and who was proficient in the English language. This ensured that the meaning was not lost. Qualitative data from the in-depth interviews were transcribed verbatim before analysis. All transcripts were uploaded into NVivo 20.7.0 [24] for analysis.

Qualitative data analysis was guided by content analysis techniques. Initially, the study team read through printed transcripts and identified emerging themes. Using the coding approach proposed by Maxwell (1996), the team generated a list of relevant codes that led to the emerging themes [25]. The transcripts were coded in NVivo. The coded text was used to generate categories of analytically meaningful data that guided the formation of themes, the interpretation of results, and the final write-up. The data were analyzed by comparing the thematic areas across different HPV self-sampling screening approaches implemented in different settings i.e., clinical and home HPV self-sampling screening approaches. The codes for each HPV self-sampling approach were created, and the codes were then grouped into categories that guided the formation of themes, as shown in the results. The query reports were generated, and relevant quotes were identified and used to support the findings.

## 3. Results

### 3.1. Demographic Characteristics of the Key Informants

The overall purposive sample was comprised of 30 participants. Of these, 24 were WLWH, 33% attained an upper primary level of education, and 33% attained secondary education. The majority (50%) of the WLWH were Catholics, and only three WLWH were Moslems. The number of WLWH for the HPV self-sampling approach was evenly distributed across home-based, clinic-based, and those that declined. The six other key informants were mainly HPV and CC service providers.

Data analysis produced mainly three themes (facilitators, barriers, and suggestions) and subthemes (motivators of HB or CB, barriers of HB or CB, and suggestions for improving HB or CB). Participants further proposed some interventions that could mitigate the impediments related to HPV self-sampling approaches. The key suggestions were mainly: considering HPV screening pre- and post-testing counseling for women attending ART clinics; improving the laboratory capacity and reducing the turnaround time for the HPV test result; and streamlining HPV logistics requisition within the National Medical Stores (NMS) supply chain. Participants further highlighted that the HPV/CC screening staff should be incentivized so that they are motivated to routinely screen WLWH, as summarized below (Table 2).

### 3.2. Facilitators for HPV Self-Sampling as a CC Screening Approach

a.Motivators for clinic-based HPV self-sampling approach

Participants identified perceived early free diagnosis and treatment, as well as education on the risks of HPV and the benefits of HPV testing, as the motivators for the clinic-based HPV self-sampling approach.

For clinic-based HPV screening, participants mentioned perceived early diagnosis and treatment of HPV as benefits of being screened at the clinic by a skilled health worker (HW), where medical instruments are housed. WLWH also believed that the HW would immediately diagnose the disease and provide treatment through the thermocoagulation process. In circumstances where a woman was found with suspicious lesions, they would be referred for further management.

“*The health workers (HWs) are capable and all the instruments available. The health workers are skilled, even if they found you complicated, they know where to refer you very fast to test for whatever they want to know. WLWH are likely to come to the clinic to screen for HPV and CC because of perceived early diagnosis and treatment*”.P1 WLWH

The key informants also mentioned being educated on HPV, the risks of HPV infection, and the benefits of HPV and VIA screening as critical motivators for the clinic-based approach. The fact that being on-site at the clinic enabled health workers to take pictures of the cervix and educate women on the state of their cervix was seen as a key motivator for clinic-based self-sampling. WLWH, through visualization and education, were able to receive knowledge on HPV and CC, as well as to return for the next appointment for screening.

“*Most of the women usually accept to screen at the clinic because they get feedback, especially those already having signs of CC. I usually take a picture of the cervix upon applying VIA and share this image with WLWH. I educate the women about HPV and CC using the images of their cervix…… WLWH are motivated to go for further management if referred because they fear progression into advanced disease*”.P26 HW

WLWH also obtained education if they had a family history of CC, where a relative or friend underwent HPV testing and CC screening or was confirmed to have CC disease. This was further evident in the quote below;

“*Recently one of my nieces sent us an audio telling us that her cervix was sick, and when diagnosed she was found with cancer of the cervix, yet she has never given birth to any child, so she was going for an operation, so there you also go and encourage other women to test because prevention is better than cure, though not everyone knows it*”.P9 WLWH

b.Motivators for home-based HPV self-sampling approach

The participants largely identified privacy during the HPV self-sampling procedure, reduced distance to return the collected sample, and a soothing HPV collection kit during the vaginal sample collection procedure as key motivators for the home-based HPV self-sampling approach. Participants echoed that the home-based approach offered privacy during the HPV DNA self-sampling procedure. WLWH mentioned that collecting the sample from home was easy and fast, as it was conducted conveniently in a private place within their homes. It was also easy to follow the sample collection procedures described by the HWs.

“*It is good because it is private, no opening up your legs for another person as they tell us to open your legs wide when being screened by a midwife at the CC clinic*.”P10 WLWH

The home-based HPV self-sampling approach reduced the distance traveled to be screened, as WLWH would pick HPV sample collection kits from the community expert clients, who were community linkage representatives (CLRs) within the community. The WLWH returned the HPV sample in its buffer to the CLRs. The community linkage representative was a selected member from the ART clinic with prior training and support in the triage of HIV patients, and they resided within the same community for the selected group of ART clients. Thus, these CLRs facilitated the distribution of HPV kits during the ART community outreach clinics and were the key contact persons for receiving and transporting the self-collected sample to the hospital laboratory. As a result of the transportation of the kits by the CLRs, sample transportation was made easy, and transport costs were reduced, as the CLRs brought the kits to the lab.

“*When women use the clinic-based HPV self-sampling approach, they usually travel a distance close to 60 kms. When samples are self-collected from and returned to the designated expert client within the community, are within reach of at least 0–10 km. The distance is easy to manoeuvre with to return the kits which are transported to the lab by the expert client*.”P25 HW

The WLWH who were in the home-based arm also mentioned that the Qvintip HPV DNA kit [30] was a fast, smooth, easy-to-use, and comfortable vaginal secretion sampling device. It was mentioned that this device was soothing during sample collection, compared to the brushes that were available at the clinic. Thus, this kit was the most preferred by WLWH randomized into the home-based arm.

### 3.3. Barriers to HPV Self-Sampling Approaches

a.All-encompassing barriers to HPV screening approach

The major barriers to HPV self-sampling using the clinic- and the home-based approaches were a lack of knowledge about HPV infection and CC disease, as well as fears related to poverty. The majority of WLWH who declined HPV self-sampling were not knowledgeable about the HPV infection and its related risks of causing CC among women. Even with group education, they preferred to be given more time to learn about HPV before they would consider being screened.

“*Women don’t make CC screening a priority, for example, those women who are always digging will only be thinking about digging and may not make their health a priority that way they will not come for testing, furthermore education status, you know if you are not educated and not knowing about the disease and its danger, they will not come for testing*.”P8 WLWH

“…*I don’t know anything, I just hear about the word cancer of the cervix, I don’t know its signs*.”P14 WLWH

Poverty was another challenge that prevented women from coming to the clinic, as they were afraid of incurring expenses of treatment should they be found with advanced CC disease. Further still, the clinic-based approach disrupted their family’s financial flow in situations where they were diagnosed with suspicious lesions, with the fear of finding the disease and the painful procedures being viewed as additional deterrents.

“…*because cervical cancer is very expensive to treat. If a person cannot afford to buy treatment, that means they will need family support, which will disorganize the family finances.*”P5 WLWH

“…*When you are very sick with a disease like cervical cancer, you may lose yourself to thoughts and there you will stop working, you get disorganized even your family members as they try to plan ways of saving your life, spend a lot of time and money*.”P8 WLWH

The failure to return the kits due to a lack of transportation was another barrier to the home-based approach, including having to travel long distances to return them to the community expert clients. Despite the community expert clients living within the communities, the distances between them were sometimes long, with additional transport costs for women to return the self-collected sample.

“*The distance within the communities is still long, like when you would go there somebody would tell you like Kikyusa yet Kikyusa is very wide with quite several villages far apart. Therefore, the distance between the woman and the expert client sometimes can be more than 10 km. The women may not have the transport fares to return the kit to the expert client.*”27 HW

b.Clinic-based HPV self-sampling barriers

At the clinic, the women reported that they faced the challenge of the intrusion of privacy. Hasty collection of the samples to create space for peers, as there was one room provided for the self-sampling approach at the clinic, was an important barrier. WLWH stated that screening with VIA pushed them to expose their private parts and made the entire screening approach uncomfortable.

“*It is very scary as the Health workers insert those instruments in you they pierce you, and you suffer pain for about four days, HPV self-sampling is very simple and not painful.*”P9 WLWH

c.Home-based HPV self-sampling barriers

Stigma and discrimination within the home-based approach were large concerns of the participants, as they were afraid to be identified as women living within the community.

“*There is a fear of discrimination and shyness because not everyone can keep a secret if I am diagnosed with cervical cancer. The lack of Family support or fear of being discriminated against stops me from seeking cancer screening services as well as collecting the sample from home*.”P13 WLWH

## 4. Discussion

This study aimed to explore the facilitators and barriers to HPV self-sampling approaches among HIV-infected women attending a rural ART clinic in Uganda. We deductively derived themes from facilitators for HPV self-sampling approaches, barriers to HPV self-sampling, and suggestions for improving HPV self-sampling among WLWH. In this discussion, we focused on findings relating to facilitators, barriers, and suggestions that would deepen our understanding of issues relating to HPV self-sampling among HIV-infected women.

### 4.1. Facilitators of HPV-Self Sampling Approaches among HIV-Infected Women

In our study, most of the respondents were Catholics by religious domination, and an equal number of participants either obtained a primary or secondary level of education. There is high dominancy in the Catholic religion of primary and secondary levels of education within Uganda, as indicated in the Uganda Demography and Health Survey 2016 [31]. Our discussion mainly focuses on the facilitators of the HPV screening approaches among HIV-infected women residing in rural areas. For some WLWH, they were motivated to screen for HPV using the clinic-based approach rather than the home-based approach because of the perceived early diagnosis and treatment in case of the suspected presence of CC disease. On the other hand, some WLWH were motivated to screen for HPV using the home-based approach rather than the clinic-based approach because of the privacy during the HPV self-sampling process. The motivators of the clinic-based HPV self-sampling were mainly considered to be the visualization of the cervix and free service, while reduced distance and the smooth sample collection kit were motivators for the home-based approach.

The findings from this work mainly focused on WLWH who were motivated to screen for HPV and CC at the clinic because of the perceived benefits of early diagnosis and treatment. Perceived benefits of the clinic-based HPV self-sampling were mainly considered to be the visualization of the cervix and free service. An example would be the findings in other studies, where HIV-infected women were motivated to screen for CC at the clinic [9], and all those who were suspected of having the disease completed treatment, compared to the HIV-negative women [28,32]. In Uganda, HIV-infected women are also motivated to be screened at the clinic and by the HW, as this increases the chances for the early diagnosis and treatment of the disease [28].

This finding was critical, similar to a systematic review conducted in Sub-Saharan Africa, where women preferred home-based HPV self-sampling because of the privacy and convenience when collecting the sample [33]. The home-based HPV self-sampling approach reduced the distance to the HIV clinic, as the samples were collected from home and returned to the expert client within the community. This finding was consistent with that of the study conducted in Cameroon, where the home-based HPV self-sampling reduced the distance to return samples to the laboratory and eased access to treatment [33]. In this study, a Qvintip kit with a smooth ‘penile’ tip was used to self-collect the sample, and it was highly appreciated by women who collected the sample from home. This result was contrary to the findings of HPV self-collected samples using cytobrushes, which women found to be irritating and sometimes broke within the vaginal canal [33].

### 4.2. Barriers to HPV Self-Sampling Approaches

The other major theme that emerged was the barriers to HPV self-sampling approaches. Barriers that cut across the two HPV self-sampling approaches were a lack of knowledge of HPV and the perceived disruption of financial flow in case CC disease was diagnosed. The barriers specific to clinic-based HPV self-sampling screening included the lack of privacy, perceived painful procedures for VIA, and a fear of finding the disease. Stigma and discrimination were reported as the major barriers to the home-based HPV self-sampling approach. Further still, the major reasons why some WLWH refused to screen were fear of finding the disease, stress, and financial disruptions related to being diagnosed with CC disease.

The other major themes that emerged were the barriers to HPV self-sampling approaches, with a lack of knowledge of HPV and the perceived disruption of financial flow in case CC disease was diagnosed being the overarching barriers. This finding was congruent with the study conducted in Ethiopia, where a lack of knowledge about cervical cancer and screening and a lack of health education on cervical cancer and HPV-based screening hindered HPV self-sampling [34]. There was barely any evidence from previous studies to ascertain the disruption of financial flow as a perceived barrier to HPV self-sampling, though the lack of finances was found to deter access to CC screening options.

Women in a similar social situation but who were HIV-negative did not prefer to be screened at clinics because of a lack of privacy [35]. The barriers specific to clinic-based HPV self-sampling screening included a lack of privacy during the screening procedures, perceived painful procedures for VIA, and fear of finding the disease. Other studies in low-income countries within the Sub-Saharan Africa region made similar findings, where embarrassment due to the lack of privacy and the fear of finding the disease deterred women from using clinic-based HPV self-sampling procedures [2,36,37]. Stigma and discrimination were reported as the major barriers to the home-based HPV self-sampling approach, as women feared that in case of a CC diagnosis, their spouses, families, and communities would discriminate against them. However, there is a paucity of data on stigma and discrimination regarding home-based HPV self-sampling; hence, more studies are required.

Further still, the major reasons why some WLWH completely refused to screen were fear of finding the disease, stress, and financial disruptions related to being diagnosed with CC disease. The decline in HPV self-sampling could be attributed to religion in this study, mainly with Muslims and Anglicans. Reasons for refusal were consistent with findings from previous studies, where the fear of finding the disease and being Moslem were the main reasons for refusing to screen for HPV and CC [28,36,37,38].

### 4.3. Suggestions from WLWH and Other Key Informants for Improved HPV Self-Sampling

Despite the identified barriers, key informants proposed some interventions to mitigate the impediments related to HPV self-sampling approaches. The key suggestions were mainly: considering HPV screening pre- and post-testing counseling for women attending ART clinics; improving the laboratory capacity and reducing the turnaround time for the HPV test results; and streamlining HPV logistics requisition within the National Medical Stores (NMS) supply chain.

a.Suggestions for clinic-based improvements

Counselling of patients within ART clinics remained amongst the key suggestions for improving the clinic-based approach to HPV self-screening. The strengthening of the lab capacity to shorten the turnaround time for HPV test results was an additional suggestion. The screening team mentioned that there was a need to streamline the cycle for the logistics distribution of VIA and HPV DNA reagents such that shortages would be controlled. Economically, it was suggested to provide extra financial support to the screening staff in the clinic to motivate them to routinely screen women for HPV. In terms of prevention, a solution proposed for reducing the risk of HPV was to consider vaccination for HPV-negative women living with HIV.

b.Suggestions for home/community-based improvements

Participants proposed the formation of community ART support groups, including a triage support system and selection criteria for community ART experts who support the community and reside in the area.

There is a lack of information on the feasibility or effectiveness of the above interventions. Thus, more studies are needed to understand the potential value of these suggestions. It was highlighted that the HPV/CC screening staff should be incentivized so that they are motivated to routinely screen WLWH. Digital financial incentivization was studied in other programs, such as immunization [39], but there is barely any evidence of how it works in CC screening projects.

## 5. Limitations

The study was prone to social desirability bias, and we did not verify the authenticity of the views shared. However, to mitigate the risk of social desirability, the objectives of the study were well explained, and we invested in building rapport at the beginning of the study.

This was mainly a qualitative study where views were collected from a small, purposively selected sample and might not provide a general representation of all WLWH in rural areas.

## 6. Conclusions

We, therefore, conclude that perceived early diagnosis for HPV and CC enhances clinic-based HPV self-sampling, while privacy enhances WLHW’s preference for the home-based HPV self-sampling approach. However, fear of finding a disease and the lack of knowledge of HPV and CC hinders HPV self-sampling. Finally, designing pre- and post-testing counseling programs in HIV care is likely to increase the demand for HPV self-sampling. It will also be required for HIV clinics to create safe spaces at ART clinics for clinic-based HPV self-sampling. There is a need to design a clear strategy for home-based HPV self-sampling while leveraging ART community linkage systems.

## Figures and Tables

**Table 1 ijerph-20-06004-t001:** Demographic characteristics of the key informants.

Sno	Participants	Age in Years	Educational Status	Religion	HPV Screening Approach
1	P1	25–35	Secondary	Catholic	CB
2	P2	25–35	Secondary	Pentecostal	HB
3	P3	25–35	Illiterate	Moslem	CB
4	P4	25–35	Tertiary	Moslem	Declined
5	P5	25–35	Illiterate	Catholic	HB
6	P6	25–35	Primary	Catholic	CB
7	P7	25–35	Primary	Anglican	Declined
8	P8	25–35	Secondary	Catholic	CB
9	P9	25–35	Secondary	Catholic	CB
10	P10	25–35	Secondary	Anglican	HB
11	P11	36–49	Secondary	Catholic	Declined
12	P12	36–49	University	Pentecostal	HB
13	P13	36–49	Illiterate	Anglican	CB
14	P14	36–49	Illiterate	Pentecostal	HB
15	P15	36–49	Primary	Catholic	HB
16	P16	36–49	Primary	Pentecostal	Declined
17	P17	36–49	Illiterate	Catholic	CB
18	P18	36–49	Secondary	Moslem	Declined
19	P19	36–49	Secondary	Anglican	CB
20	P20	36–49	Illiterate	Catholic	HB
21	P21	36–49	Primary	Anglican	Declined
22	P22	36–49	Primary	Catholic	CB
23	P23	36–49	Primary	Catholic	CB
24	P24	36–49	Primary	Catholic	HB
Other informants					
25	P25 (HW)	36–49 Male	University	Anglican	N/A
26	P26 (HW)	25–35 Female	Tertiary	Catholic	N/A
27	P27 (HW)	36–49 Male	University	Anglican	N/A
28	P28 (MOH)	36–49 Male	University	Catholic	N/A
29	P29 (UCI)	25–35 Female	University	Anglican	N/A
30	P30 (HW	36–49 Female	University	Catholic	N/A

HW: health worker; MOH: Ministry of Health; UCI: Uganda Cancer Institute; HB: home-based HPV self-sampling; CB: clinic-based HPV self-sampling; N/A: not applicable.

**Table 2 ijerph-20-06004-t002:** The facilitators, barriers, and suggestions for HPV self-sampling approaches among WHLH.

Codes	Subthemes	Themes	Major Source Documents
Perceived early free diagnosis and treatment. Education on the risks of HPV Benefits of HPV testing	Motivators for clinic-based HPV self-sampling	Facilitators for HPV self-sampling	[26,27,28]
Privacy during the HPV self-sampling procedure Reduced distance to return the collected sample Soothing HPV collection kit during the vaginal sample collection procedure	2. Motivators for home-based HPV self-sampling approach		
Lack of knowledge of HPV and CC disease Fears related to poverty. Fear of finding disease Lack of transport Failure to return the sample	3. All-encompassing barriers to the HPV screening approach.	Barriers to HPV self-sampling approaches	[1,29]
Lack of privacy Perceived pain related to the procedure	4. Clinic-based HPV self-sampling barriers		
Stigma Discrimination Lack of family support	5. Home-based HPV self-sampling barriers		
Provide a comprehensive HPV test counselling. Reduce turnaround time for HPV DNA results. Streamline logistics distribution. Vaccination	6. Suggestions for clinic-based improvements	Suggestions for improved HPV self-sampling approaches	[1,27,29]
Use of community ART groups for HPV self-sampling	7. Suggestions for home-based improvements		

## Data Availability

The data from this study are the property of Makerere University and the University of KwaZulu-Natal. All interested readers can request the data from Makerere University, School of Public Health through the principal investigator, Agnes Nyabigambo, Email: anyabigambo@musph.ac.ug | Mobile Phone: +256774135496; and the BIOMEDICAL RESEARCH ETHICS ADMINISTRATION Research Office, Westville Campus, Govan Mbeki Building University of KwaZulu-Natal P/Bag X54001, Durban, 4000 KwaZulu-Natal, South Africa Tel.: +27-31-260-4769 Fax: +27-31-260-4609, Email: BREC@ukzn.ac.za.

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
