# Peer review of "Facilitators and Barriers to HPV Self-Sampling as a Cervical Cancer Screening Option among Women Living with HIV in Rural Uganda"

_ijerph, 2023, doi:10.3390/ijerph20116004_

Round 1
Reviewer 1 Report
Cervical cancer is a significant public health challenge for women globally, with a higher end burden experienced by those living with HIV. Human papillomavirus (HPV) self-sampling has been proposed as a potential cervical cancer screening option for women, especially in low-resource settings, where access to healthcare services is limited. This scientific research paper titled "Facilitators and Barriers to HPV Self-Sampling as a Cervical Cancer Screening Option among Women Living with HIV in Rural Uganda" investigates the facilitators and barriers of HPV self-sampling as a screening tool for cervical cancer among women living with HIV in rural Uganda. Although the paper highlights the facilitators and barriers to HPV self-sampling efficaciously, there are few concerns, and scope for betterment of the manuscript.
1. Though the burden of HPV is huge in women living with HIV and there is a clear need for the enhanced number of screenings for HPV in WLWH, how could “facilitators” and “barriers” for cervical cancer self-sampling in WLWH be different for the general (random) women of Uganda, who face almost similar social situation? Is there a difference in facilitators and barriers, and if so, why there is a difference? The answer to this can take a place in the discussion section of the paper.
2. Line 47-48: “ ….infection and an estimated 3.6% of the Ugandan woman population harbours HPV 16/18 at any given point in time” – The line needs to be rephrased to avoid suggesting that the 3.6% prevalence is a fixed number at any time
3. Line 56-58: “WHO recommends the HPV test as a screening test and visual inspection of the cervix with acetic acid (VIA) as a triage test for women who test HPV positive” Kindly rephrase it to make it morecomprehensible.
4. Section – Research design: The section is poorly written, and the research design is not conveyed to the reader. For instance, “In this study, women were initially educated about the HPV self-sampling methods using the Vignettes Technique, collected the sample and interviewed about their experiences with the screening approach.” Vignettes technique involves briefing a hypothetical situation, where the respondent is asked to make a choice. Post that, for the sample collection, did the study offer them the liberty of choosing their preferred method? And how home-based sample collection was designed? Were participants' opinions solely based on this study, or were they from their previous experiences? The entire section needs re-writing to clearly specify the design of the study.
5. Line 139-141: “The team generated a list of relevant codes that lead to the emerging themes as proposed by Maxwell (1996) (25)” The cited paper has no relevance to the presented data. Please provide the correct reference citation.
6. The relevance of including the "religion" demographic in the study is never discussed anywhere in the manuscript and needs clarification.
7. Table 2. Separator line between “barriers to HPV self-sampling approaches” and “suggestions for improved HPV self-sampling approaches” is missing.
8. Line 177-182: The line needs to be rephrased with proper usage of grammar and punctuation marks to make it comprehensible.
9. Line 244-245: “Thus, this kit was most preferred by WLWH, randomized into the home-based arm.” The statement regarding the preferred kit needs clarification. Is this conclusion being from the present study? Because there is no mention of this kit or any other kit in the methods section. In that case, what all other kits were included in the study to say WLWH “preferred” this particular kit. If the conclusion is from other study, a proper reference should be cited.
10. Section: a. All-encompassing barriers to HPV screening approach: Why any statement/quote from the subjects who declined the HPV screening approach is not included in the section? Opinion from these subjects can justify the factual barrier to HPV sampling approach.
11. Line 282-285: Translation is poorly done, and the lines are difficult to understand. For instance, “like when you would go there, like somebody would tell you like Kikyusa yet Kikyusa is very wide” looks like it is translated word-to-word than translating it in a way to better convey the meaning.
12. Line 294-296: ““I experience pain in the cervix and the vagina. It is very scary, they insert those instruments in you they pierce you, and you suffer pain for about four days, yet this one (clinic-based HPV self-sampling) is very simple and not painful” This statement can be confusing. Kindly mention in the study design section that clinic-based HPV self-sampling was performed as analogy to home based self-sampling. Also, the term “clinic-based” in the parenthesis cab be removed as stating self-sampling is sufficient.
13. The entire discussion part has a lot of space for improvement. The discussion section of the current manuscript is merely repeating the same facts from the result section (multiple times within discussion section, as well). The authors should provide a critical evaluation of the study results and more reference should be included so as to know of the contemporary situation in similar studies. Also, the result of the study could be discussed in a way explaining the possible reasons for the kind of outcome it has demonstrated. For instance, the section is not discussing the relation of the educational status with the out come of the study. The section can also link the outcome with the women sex education status of Uganda, giving a proper reference.
14. Please check and correct the citation format for reference number 30. (Line-530)
15. The whole paper has lot of scope for the correction in terms of grammatical aspects, right use of punctuations and articles.
Author Response
See the point by point response attached.

Reviewer 2 Report
In the current study, the authors tried to analyze the factors that could affect HPV self-sampling in women with HIV infection. The study is well written but I have some comments
1- Many abbreviations were mentioned in the abstract such as WLWH and CC without mentioning the full name. Kindly write the full name when the abbreviation is first mentioned.
2- In the introduction section, put more focus on the correlation between CC, HPV, and HIV.
3- In the discussion section, the first 3 paragraphs are just a repetition of already mentioned data. Please focus more on comparing your findings with other reports that were performed in other African countries (similar cultures) and other countries around the world (different cultures).
Author Response

(The authors gave the same response as above.)

Reviewer 3 Report
In this study by Nyabigambo et al, the authors explore the facilitators and barriers to HPV self-sampling as a cervical cancer screening approach among HIV infected women living in rural Uganda. The authors used a health promotion model in 24 participants and analyzed the motivators and barriers to self-sampling and clinic-based sampling. Some of the facilitators for the self-sampling approach included privacy, motivators for the clinic-based approach included the visualization of the cervix. Some of the barriers to self-sampling included lack of knowledge of HPV, for clinic-based sampling included lack of privacy, fear of finding disease. I believe this study is interesting since its on an underserved population with risk factors for HPV infection (HIV), and due to the fact that HPV self-sampling is an emerging and innovative technique.
Some issues I believed should be addressed are the following:
-The discussion is between sections b. suggestions for home/community-based improvements (line 322) and facilitators of HPV-self sampling approaches among HIV infected women (line 360). Consider moving the discussion to before the conclusion (line 426) since this affects the flow of how the study is read.
-The authors could expand or mention the number of participants that shared a certain view,
-One limitation is the sample size; authors could address this in the limitation section or the rationale behind the sample size chosen for this study.
-Authors could include the in-depth interview as a supplementary material
Author Response
See the point-by-point response attached.

Reviewer 4 Report
See attached file

Author Response

(The authors gave the same response as above.)

Round 2
Reviewer 1 Report
Thank you for addressing all the raised concerns.